# The Impact of Street Tree Height on PM2.5 Concentration in Street Canyons: A Simulation Study

**Junyou Liu** **, Bohong Zheng \*, Yanfen Xiang and Jia Fan**

School of Architecture and Art, Central South University, Changsha 410083, China
* Correspondence: zhengbohong@csu.edu.cn

**Abstract:** With the rapid development of cities and the rapid increase in automobile ownership, traffic has become one of the main sources of PM2.5 pollution, which can be reduced by road greening through sedimentation, blocking, adhesion, and absorption. Using the method of combining field monitoring and ENVI-met simulation, the influence of the tree height on the PM2.5 concentration on both sides of the city streets was discussed. The influence of tree height on PM2.5 under five conditions was analyzed, including 10 m tall trees (i), 15 m tall trees (ii), alternating distribution of 15 and 10 m tall trees (iii), 5 m tall trees (iv), no trees on either side of the road (v). The results show that: Roadside trees can increase the concentration of PM2.5 in the narrow space of street canyons. However, without roadside trees, PM2.5 from traffic sources is not reduced in time, it is more easily spread to the distance. When the height of the roadside trees is 5 m and their crown widths are smaller than those of other trees, there is a relatively wide space between them. Compared with the higher roadside tree models with larger crown widths, the concentration of PM2.5 on the roadway and the downwind sidewalk is relatively low. In the three models (i–iii) with tree height above or equal to 10 m, the PM2.5 concentration around the trees do not show regular change with the change in tree height. Due to the tree height of 10 and 15 m, the crown width is large enough, and the alternate distribution of tree height of 15 and 10 m fails to make the PM2.5 concentration in the models lower than the models with tree height of 15 m or 10 m. The reasonable height of roadside trees in street canyons helps improve the wind circulation to promote the diffusion of PM2.5 pollution. There is no optimal height of roadside trees for PM2.5 pollution in street canyons, thus it is necessary to select the height reasonably according to the specific situation.

**Keywords:** PM2.5; street canyon; street tree; tree height; deposition and sedimentation

## 1. Introduction

In this increasingly globalized international community, countries around the world have paid considerably more attention to the significance of achieving the Sustainable Development Goals (SDGs) of the United Nations and committed to achieving these goals. Air pollution control is an important part of the Sustainable Development Goals of the United Nations. One of the important elements of Sustainable Development Goal 3.9 (SDG 3.9) is to greatly reduce the number of human diseases and deaths caused by air pollution worldwide by 2030 [1]. Fine particulate matter (PM2.5) in the atmosphere is one of the main urban pollutants and originates from inadequate combustion of fossil fuels, energy production, residential energy use, exhaust emissions from transportations, fertilizer use, combustion of agricultural wastes, forest fires, solvent use, industrial production, and wind-blown mineral dust from arid areas [2–4]. Due to the fact that its diameter is less than 2.5 μm, it can enter the pulmonary alveoli and harm the human body. A great number of studies have proven that the PM2.5 concentration is positively correlated with the incidence rate of cardiovascular and respiratory diseases [5]. Moreover, it is believed that meteorological conditions (such as temperature, wind direction, wind speed, relative humidity, rainfall, and snowfall) are related to the PM2.5 concentration [6,7]. Furthermore,

as one of the main pollutants in the atmosphere, PM2.5 not only has negative impacts on human health, but also poses serious threats to the sustainable development of society. In the Air Quality Guidelines issued by the World Health Organization (WHO) in 2021, the target values of the annual average PM2.5 concentration and the daily average PM2.5 concentration were adjusted from 10 and 25 $\mu g/m^3$ in the guidelines issued by the WHO in 2005 to 5 and 15 $\mu g/m^3$, respectively [8]. The lowering of the standards not only indicates that the WHO has higher requirements for the control of atmospheric PM2.5 pollutants, but also indicates that different countries around the world have to achieve a higher target and continue to work hard to create a good air environment for the residents.

PM2.5 from the traffic is one of the main sources of urban atmospheric fine particulates, mainly from the exhaust gases released by fuel combustion, road dust, and dust released by road construction [9]. There is no doubt that exploring a logical way to reduce PM2.5 pollution from traffic is beneficial for achieving SDG 3.9 and promoting the sustainable development of human society. Since PM2.5 from the traffic is mainly from the above-mentioned three sources, when comparing different modes of transportation, the PM2.5 concentration around people in an open environment (while walking or cycling) may be higher than an enclosed environment (private car, bus or subway). Similarly, if the route of cyclists and pedestrians is surrounded by traffic stopping at or crossing the sidewalk, they may be exposed to high concentrations of PM2.5 [10]. If the window of a private car is opened for ventilation while driving, PM2.5 outside the car will accumulate in the vehicle, the air conditioner of which can purify the air inside the car and reduce PM2.5 that entered from outside the car. PM2.5 not only seriously affects the outdoor environment, but also seriously affects the environment inside the compartment. Some researchers found that the PM2.5 concentration in the compartment of a bus was higher than the external environment [11–13].

Urban roads have a large traffic flow and a large amount of dust, but the green belt can retain the dust. In some studies, it was found that an urban form would affect the dust retention capacity of leaves. Factors related to urban morphology include building packing density, street aspect ratio, street width-to-length ratio, and roof geometry [14,15]. The differences in the form of a neighborhood may lead to a different distribution of PM2.5 in street canyons and therefore, the effect of plants on the surrounding PM2.5 concentration will also be different. The influence of relevant meteorological factors should be sufficiently considered while considering those of plants on PM2.5 in street canyons. Irga et al. (2016) [16] conducted a study of air quality conditions at a few different sites in Sydney and found that sites with less green space had a higher concentration of aerosolized particles. Some researchers found in studies that road greening caused an increase in the PM2.5 concentration on highways and a decrease in PM2.5 concentration on the lane and sidewalks of non-motorized vehicles [17]. Some researchers compared and analyzed the PM2.5 retention capacity of different plants and configuration structures in the green belt of the road. For example, Chen et al. (2022) [15] found that the PM2.5 retention capacity of the plants from high to low were: Trees and shrubs, pure forest of shrubs, trees with grass and shrubs, shrub grass, trees with shrubs, and pure forest of trees. Among the more than ten plant species studied by the researchers, *Ophiopogon japonicus* and *Podocarpus macrophyllus* have higher PM2.5 retention capacity per unit leaf area. On a single plant scale, *Photinia serrulata* and *P. macrophyllus* have high dust retention capacity, while *Oxalis corniculata* has low dust retention capacity. In terms of PM2.5 retention capacity of plants per unit area, *P. macrophyllus* has a high dust retention capacity, while *O. corniculata* and *Magnolia grandiflora* have low dust retention capacity. Some researchers investigated the optimal layout from the perspective of the layout of road greening to minimize the concentration of aerosol particles on the sidewalk. In the study, six urban street models were selected, including no greenery, two rows of trees, three different types of trees in three rows, and four rows of trees. The study found that the model with the lowest mass concentration of aerosol particles on the sidewalk was three rows of trees of different heights, and the

model with the lowest concentration of aerosol particles was four rows of trees of the same height [14].

Road greening does not always have a beneficial effect on reducing the ambient PM2.5 concentrations. Some researchers around the world have found that border trees have negative effects on ambient air quality [18,19]. They attributed this to the fact that border trees can reduce the ventilation capacity of street canyons and thus, increase the mean concentration of pollutants [20–22]. Chávez-García and González-Méndez (2021) [23] pointed out that the spatial pattern of plants and their leaf surface features were two important factors determining their effect on air quality. Buccolieri et al. (2018) [24] indicated in their study that from a purely dynamical point of view, border trees had different rules when it comes to affecting the PM2.5 concentrations in different wind directions. They increase the ambient PM2.5 concentration when the wind direction is perpendicular to the streets, and decrease the ambient PM2.5 concentration when the wind direction is parallel to the streets. However, in real life, the effect of border trees on PM2.5 concentrations will produce different results based on different complex situations at different sites in different wind directions.

There are currently few studies on the different influences of different heights of plants in the green belts on the roads on the PM2.5 concentration on the sidewalk. This study is based on a section of Shaoshan South Road, a main road in the north-south direction in the high-density central urban area of Changsha City (the capital of Hunan Province in China, N27°51′ to N28°41′ and E111°53′ to E114°15′) to construct a study model. This section has six roadways, with high-rise residential buildings and multi-story commercial shops distributed on either side. Moreover, it has characteristics similar to many road sections in the central urban area and therefore, was selected as a representative section for research purposes. In this paper, a model was built based on the segment of Shaoshan Road, a main street in Changsha, and the ENVI-met software was used for modeling and simulation analysis. Through a comparative analysis of the influences of different heights of plants in the green belts of the roads in Changsha (the main pollutant in the atmosphere being PM2.5) on the PM2.5 concentration on the sidewalk, this paper aims to provide a scientific theoretical basis for proposing strategies to reduce the PM2.5 concentration on the sidewalk from the perspective of the height of the green belts on the roads.

## 2. Materials and Methods

### 2.1. ENVI-Met

ENVI-met is a three-dimensional urban microclimate simulation software based on thermodynamic principles and computational fluid mechanics (CFD) used for small and mesoscale applications, developed by German scholars Michael Bruse et al. [25]. The three-dimensional non-static hydrology model can be used to simulate the interaction between the atmosphere, plants, and surfaces of buildings. The predicted values and spatial distribution of environmental factors, such as pollutant concentration, wind speed, temperature, and humidity can be the output, realizing the dynamic coupling analysis.

### 2.2. Introduction to the Principle of Pollutant Diffusion Simulation

Envi-met adopts the standard convection-diffusion equation proposed by Bruse (2007) [26] to simulate the diffusion of exhaust and pollutants:

$$\frac{\partial \chi}{\partial t} + u\frac{\partial \chi}{\partial x} + v\frac{\partial \chi}{\partial y} + w\frac{\partial \chi}{\partial z} = \frac{\partial}{\partial x}\left(K_x\frac{\partial \chi}{\partial x}\right) + \frac{\partial}{\partial y}\left(K_x\frac{\partial \chi}{\partial y}\right) + \frac{\partial}{\partial z}\left(K_x\frac{\partial \chi}{\partial z}\right) + Q_x(x,y,z) + S_x(x,y,z) \tag{1}$$

where $\chi$ refers to the composition of exhaust and particulate matter in the simulated atmosphere, in $\left[\mathrm{mg}(\chi)\mathrm{kg}^{-1}\,(\mathrm{air})\right]$; $u$ refers to the wind speed; $v_{s/d}$ refers to the sedimentation speed; k refers to the Boltzmann constant ($=1.38 \times 10^{-23}$ JK$^{-1}$); $\Delta w$ refers to the distance from the center of the grid to the solid surface; $\frac{\partial \chi}{\partial t}$, $\frac{\partial \chi}{\partial x}$, $\frac{\partial \chi}{\partial y}$, $\frac{\partial \chi}{\partial z}$ refer to the differential of exhaust and particulate matter $\chi$ to t, x, y, z; $\frac{\partial}{\partial x}$, $\frac{\partial}{\partial y}$, $\frac{\partial}{\partial z}$ refer to the deriva-

tives of respective x, y, z; $Q_x$ and $S_x$ refer to the pollution source and sedimentation type (sedimentation or chemical reaction); $Q_x$ is in $\left[ \text{mgkg}^{-1}\text{s}^{-1} \right]$.

The pollution source $Q_x$ can be calculated using the following formula:

$$Q_x(x, \ y, \ z) = q^* \cdot (vol \cdot \rho)^{-1} \tag{2}$$

where

$$q^* = \ ql \cdot \Delta x, \ y \tag{3}$$

The pollution source $Q_x$ can be classified into point source, line source, areal source, and volume source. The traffic pollution source belongs to the line source, and its release rate is usually expressed by ql in $\left[ \text{mgs}^{-1}\text{m}^{-1} \right]$. For the convenience of calculation, $q^* [\text{mgs}^{-1}]$ is calculated according to the release rate. The calculation formula is (2). Vol represents the air volume in $\left[ \text{mgm}^{-3} \right]$.

Deposition or the chemical reaction $S_x$ can be expressed using the following formula:

$$S_x(x, \ y, \ z) = \chi^{\downarrow}(z) + \ \chi_{\downarrow}(z) + \ \chi_{\text{plant}(z)} \tag{4}$$

where

$$\chi^{\downarrow} (z) = v_{s/d} \frac{\chi(z + 1)}{\Delta z} \tag{5}$$

$$\chi_{\downarrow}(z) = -v_{\frac{s}{d}} \frac{\chi(z)}{\Delta z} \tag{6}$$

$$\chi_{\text{plant}(z)} = \text{LAD}(x, \ y, \ z) \cdot v_{d,p} \cdot \chi(z) \tag{7}$$

$\chi^{\downarrow} (z)$ refers to the gain of particles sedimented from the upper layer; $\chi_{\downarrow}(z)$ refers to the downward flux of particles per unit time due to settlement under gravity; the total particle loss caused by blade surface deposition is expressed by $\chi_{\text{plant}(z)}$; $v_{s/d}$ refers to the sedimentation speed; $\Delta z$ refers to the vertical distance between the predicted calculation points of horizontal Z and z + 1.

The concentration change caused by deposition and sedimentation can be expressed by

$$\frac{\partial \chi}{\partial t}\bigg|_{sed} = v_{s/d} \frac{\partial \chi}{\partial z} \tag{8}$$

where $v_{s/d}$ refers to the sedimentation speed; $\frac{\partial \chi}{\partial z}$ refers to the differential of pollution $\chi$ to z.

According to the relevant standard convection-diffusion equation, pollutant type, release rate, particle sedimentation speed, other related chemical reactions, Boltzmann constant related to temperature and energy, wind speed, and the influence of blade surface on particles are fully considered in the simulation by ENVI-met software of exhaust and pollutants diffusion.

### *2.3. Model Building*
### 2.3.1. Land Surface and Buildings

The model was built based on a segment of the main street located at Shaoshan South Road, Tianxin District, Changsha. According to the specific information of this road section, the researchers first built a three-dimensional model on SketchUp 2021, as shown in Figure 1. They input material information to the building and ground using the INX plug-in in SketchUp, and converted the model into a model that could be used in ENVI-met software.

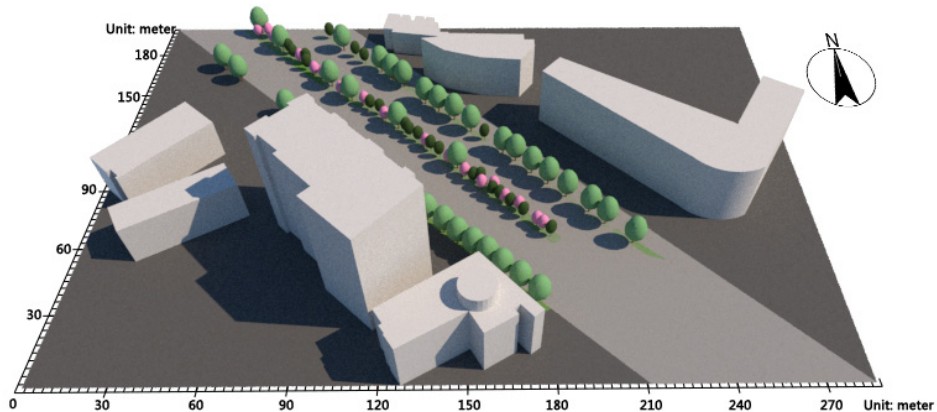

**Figure 1.** Spatial model of study area.

2.3.2. Vegetations

In this study, ENVI-met 5.0.3 was used for modeling and simulation analysis. In this simulation, in the horizontal level, the number of grids of the model was set to $136 \times 107$, the size of each grid was $3 \text{ m} \times 3 \text{ m}$, and the total area size was $408 \text{ m} \times 321 \text{ m}$. Since the model was built with sufficient space between the building and the model boundary, the researchers did not add the nested grid around the model.

In terms of horizontal range area, this study area is similar to some existing studies, and there are no doubts about the feasibility of this scale [27]. Border trees in the central urban area of Changsha as well as the studied area in this research are mainly camphor trees. Researchers constructed 3D models of camphor trees in ENVI-met. There are also some other plants in this research. All plants and their relevant parameter settings are listed in the Table 1 below. To control for the influence of different crown shapes on the study, all camphor trees of different heights were uniformly set to have spherical crowns and medium-density leaves. This was most similar to the crown information observed by researchers in this road section.

**Table 1.** Various plants and relevant parameters of the studied road section.

| Vegetation | Plant Height | Crown Width (m) | LAD (m$^2$/m$^3$) |
|---|---|---|---|
| | 5 m | 3 | 1.80 |
| *Cinnamomum camphora* | 10 m | 7 | 1.80 |
| | 15 m | 11 | 1.80 |
| *Magnolia grandiflora* | 8 m | 5 | 1.60 |
| *Prunus cerasifera* | 7 m | 5 | 1.80 |
| *Chinese photinia* | 1 m | 1 | 2.50 |
| *Ophiopogon japonicus* | 15 cm | - | 0.30 |

In this study, ENVI-met Model 1 was built according to actual conditions. In practice, the height of the trees on both sides of the road is about 10 m. In this study, the height of the trees on both sides of the road was changed, and the model was built with the alternating distribution of 15 and 10 m tall trees (Model 2), 15 m tall trees (Model 3), 5 m tall shrubs (Model 4), and no street greenery (Model 5). In this study, the same observation points in the surrounding area of the road were selected to measure the PM2.5 concentration on the observation point in different time periods. In the five models, the green belt in the middle of the road always corresponds to the actual conditions. This could help in controlling the height of vegetation in the roadside green belt as a single variable to analyze the influence of different heights of roadside vegetation on the PM2.5 concentration in the surrounding area.

### 2.3.3. Meteorological Condition

Through field measurement, data on hourly air temperature, humidity, wind speed, and wind direction between 13:00 and 23:00 on 24 August 2022 in this study area were obtained, using Kestrel 5500 handheld weather meter produced by Kestrel from the United States as a measuring equipment, and the instrument precision is shown in Table 2 below. Average values of the hourly data obtained would be entered into ENVI-met for simulation. Hourly meteorological data are shown in Table 3 below.

**Table 2.** A description of precision, resolution, and range of Kestrel 5500 handheld weather meter.

| Sensor | Accuracy (+/−) | Resolution | Specification Range |
|---|---|---|---|
| Ambient temperature | 0.5 °C | 0.1 °C | −29.0 to 70 °C |
| Relative humidity | 2% RH | 0.1% RH | 10 to 90% 25 °C non-condensing |
| Wind speed | Larger than 3% of reading, least significant digit or 20 ft/min | 0.1 m/s | 0.6–40 m/s |
| Compass | 5° | 1° 1/16 Cardinal Scale | 0 to 360° |

**Table 3.** Hourly meteorological data measured in the study area.

| Date | Time | Ambient Temperature (°C) | Relative Humidity (%) | Wind Speed (km/h) | Prevailing Wind Direction (°) |
|---|---|---|---|---|---|
| 24 August 2022 | 13:00–14:00 | 31.7 | 61.8 | 5.1 | 232.0 |
| | 14:00–15:00 | 32.9 | 59.7 | 5.3 | 274.6 |
| | 15:00–16:00 | 33.0 | 59.7 | 6.6 | 255.8 |
| | 16:00–17:00 | 32.8 | 61.2 | 6.2 | 263.1 |
| | 17:00–18:00 | 32.7 | 62.7 | 5.1 | 256.7 |
| | 18:00–19:00 | 32.8 | 63.4 | 4.4 | 276.6 |
| | 19:00–20:00 | 32.2 | 66.7 | 4.8 | 282.3 |
| | 20:00–21:00 | 32.0 | 67.8 | 3.7 | 251.0 |
| | 21:00–22:00 | 31.4 | 71.4 | 3.1 | 291.6 |
| | 22:00–23:00 | 31.2 | 72.2 | 4.9 | 247.1 |

### 2.3.4. PM2.5 Concentration

For the estimation of the hourly traffic volume on the road section from 13:00 to 23:00 on 21 August, the researchers used a mechanical counter to count the traffic volume within 10 min in each hour and estimated the traffic volume within 1 h by the measured traffic volume within 10 min. Since the studied road had six lanes in two directions, it was difficult for the researchers to count traffic flow in both directions at the same time. Therefore, they counted the traffic flow in 10 min in the lanes from south to north and from north to south, respectively, and estimated the traffic flow in each hour. The total traffic flow in this segment in 1 h was the sum of the two. Table 4 below shows the statistical results.

To reduce vehicle emissions, the Chinese central government has introduced a series of emission limits for vehicles in stages. Changsha Municipal Government implemented the National IV, V, and VI standards in 2011, 2017, and 2020, respectively [28,29]. After the implementation of National IV, V, and VI standards, new vehicles must meet the relevant standards before applying for vehicle license plates and usage on the road. To estimate the PM2.5 emission coefficient of vehicles, the researchers referred to the limits for PM2.5 emitted by vehicles in the National IV, V, and VI standards. In addition, in recent

years, some studies have estimated the PM2.5 emission coefficients of different types of vehicles [30–32]. According to the related materials and a specific analysis of the study area, the researchers finally determined the emission coefficients of all types of vehicles in the simulation study, as shown in Table 5 below.

**Table 4.** Traffic flow statistics of the studied road section.

| Date | Time | Traffic Flow in 10 min on the Road from North to South | Traffic Flow in 10 min on the Road from South to North | Estimated Traffic Flow on the Road from North to South in 1 h | Estimated Traffic Flow on the Road from South to North in 1 h | Estimated Value of Total Traffic Flow within 1 h |
|---|---|---|---|---|---|---|
| 24 August 2022 | 13:00–14:00 | 400 | 371 | 2400 | 2226 | 4626 |
| | 14:00–15:00 | 407 | 478 | 2442 | 2868 | 5310 |
| | 15:00–16:00 | 431 | 448 | 2586 | 2688 | 5274 |
| | 16:00–17:00 | 453 | 402 | 2718 | 2412 | 5130 |
| | 17:00–18:00 | 473 | 461 | 2838 | 2766 | 5604 |
| | 18:00–19:00 | 495 | 459 | 2970 | 2754 | 5724 |
| | 19:00–20:00 | 465 | 430 | 2790 | 2580 | 5370 |
| | 20:00–21:00 | 406 | 376 | 2436 | 2256 | 4692 |
| | 21:00–22:00 | 409 | 363 | 2454 | 2178 | 4632 |
| | 22:00–23:00 | 344 | 301 | 2064 | 1806 | 3870 |

**Table 5.** Emission coefficient for different types of motor vehicles.

| Vehicle Type | Passenger Car | Light Duty Vehicle | Heavy Duty Vehicle | Bus | Coach |
|---|---|---|---|---|---|
| Emission factor (mg/km·vehicle) | 6 | 1.9 | 44 | 44 | 44 |

### 2.3.5. Data Input and Simulated Operation

After building five different scenario models of the height of street trees on two sides and collecting all the associated meteorological data and PM2.5 pollution concentration data, the researchers input data into the ENVI-guide module in ENVI-met, and performed a simulation. Since it was only allowed to input a fixed wind speed and wind direction before each simulation operation in ENVI-guide, in this study, the time of each simulation was set to 1 h to set different wind directions and wind speeds according to the actual conditions in each hour. In Model 1 with a height of street trees on both sides of 10 m, in this study, the hourly PM2.5 pollution concentration simulation was performed from 13:00 to 23:00 on 24 August for a total of 10 times. As Model 1 was built according to the actual conditions of the road section, the simulated value of PM2.5 concentration at the coordinate point (58, 50) in Model 1 and the hourly PM2.5 pollution concentration value measured by researchers at the measured points corresponding to the coordinate points would be used to analyze experimental errors. In addition, the researchers input the corresponding measured data using Models 2 to 5 built by ENVI-guide, and performed the simulation of the PM2.5 pollution concentration of different models in the same period. The temperature, humidity, wind direction, and wind speed in the simulation were derived from hourly average data measured by researchers using the Kestrel 5500 on the same day. The background concentration of PM2.5 input into the simulation process was derived from the average concentration of PM2.5 over the previous hour as measured by researchers using TSI 8530.

### 2.4. Statement on Rationality of Simulation Study and Analysis of Simulation Error

Some scholars have already explored the reliability of ENVI-met simulated aerosolized particles. Some of their findings have shown high accuracy of the simulation study, while others indicated that the actual situation was significantly under- or overpredicted in the simulated data. For example, Paas and Schneider (2016) [27] compared the measured data of PM10 in four specific study areas at urban parks of two cities in Germany to the corresponding ENVI-met-simulated data, and found that the actual PM10 concentration in the ENVI-met-simulated data was significantly underpredicted (fractional bias: 1.46–1.8). On the other hand, Hofman and Samson (2014) [33] simulated PM10 in urban street canyons using ENVI-met, and they pointed out that ENVI-met performed well in the simulation through calculations of the correlation between simulated and actual values, with r = 0.58–0.79 and RMSE = 74–102% in the first half of the street canyon and r = 0.58–0.64 and RMSE = 74–102% in the second half of the street canyon. Sun et al. (2021) [34] simulated the PM2.5 concentration at an urban road intersection and compared the simulated value to the measured value. The results of some statistical indexes are shown as follows. Absolute bias, fractional bias (FB), root mean square error, and Spearman's coefficient are 18.4, 0.42, 20.8, and 0.9266, respectively.

According to the related error analysis results, ENVI-met simulation results generally show different errors under different simulation conditions in previous relevant studies. Researchers consider it very important to conduct an error analysis in this study, which will help in assessing to what extent simulation results can reflect the actual situation. Spearman's correlation coefficient is a common method for measuring the dependence of two variables, whereas fractional bias can be used to compare inconsistencies between the samples [35]. In this study, Spearman's correlation coefficient was used to explore the correlation between simulated samples and actual situations, and to explore the errors present in this study. Meanwhile, the average absolute error was used to explain the extent to which samples over- or underpredict the actual situation.

The error analysis in this study was mainly based on the measured and simulated values of PM2.5 concentration at a specific location in the study area between 13:00 on 24 August 2022 and 23:00 on the same day. It was indicated above that the simulation of PM2.5 concentration mainly depended on ENVI-met. TSI 8530 aerosol monitors manufactured by TSI of the United States were used for this study to measure PM2.5 concentrations at a few specific locations, with some details of TSI 8530 aerosol monitors listed in Table 6 below. As part of the study, researchers used the instrument to measure hourly PM2.5 concentrations at a measuring site for 10 h on 24 August 2022 from 13:00 to 23:00. According to the measuring sites, the simulated values of PM2.5 concentration at corresponding points and in corresponding time periods in the diagram of simulation results were found. In addition, based on the study site, the researchers calculated the errors between the measured and simulated values by Spearman's correlation coefficient and average absolute error.

**Table 6.** Related technical parameters of TSI 8530.

| Facility | TSI DustTrak II Aerosol Monitor 8530 |
|---|---|
| Range | 0.001 to 400 mg/m$^3$ |
| Resolution | $\pm$0.002 mg/m$^3$ |
| Particle size range | Approximate 0.1 to 10 $\mu$m |
| Operational temperature | 0 to 50 °C |
| Operational Humidity | 0–95% RH, non-condensing |

## 3. Results and Discussion

As sustainable development gains increasing attention around the world, pollution control of PM2.5 has become a crucial environmental protection issue in China. As a rapidly developing country, China is also considered one of the countries which suffer from serious PM2.5 pollution [36]. The Chinese government has paid close attention to the importance of abating PM2.5 pollution for the sustainable development of the ecological environment,

and China has become one of the first developing countries to carry out large-scale PM2.5 pollution abatement [37]. Since 2012, China has comprehensively built nationwide real-time PM2.5 monitoring stations. From that time, real-time monitoring data of PM2.5 pollution have been widely used for PM2.5 pollution prevention and control. In addition, from 2012 to the present, China has achieved remarkable results in abating PM2.5 pollution. In the first half of 2022, the average PM2.5 concentration in China's 339 cities at prefecture level and above was 32 µg/m$^3$.

In China, research into reducing PM2.5 concentration has great significance for improving people's living environment and realizing the sustainable development of society. It can be seen from Figure 2 below that the average annual PM2.5 concentration in China showed a steady reduction from 2013 to 2021. By 2021, the average annual PM2.5 concentration in China was lower than the Level-2 limit of the PM2.5 concentration of 35 µg/m$^3$, which is still higher than the guideline value of the PM2.5 concentration of 5 µg/m$^3$ of the WHO. To respond to the state's call for PM2.5 pollution abatement and meet the national goal of reaching a PM2.5 pollution between 15 and 25 µg/m$^3$, which is the lower limit, among most developed countries by 2035 [38], the Changsha Municipal People's Government proposed in the"13th Five-Year" Ecological Construction and Environmental Protection Plan of Changsha City (2016–2020) that Changsha's annual average PM2.5 concentration in 2020 should be reduced by 18% in 2015 [39]. It was proposed in Changsha's 14th Five-Year Plan (2020–2025) that the PM2.5 concentration in the city should reach 37 µg/m$^3$ by 2025 [40]. In addition, the support of relevant policies have strengthened the guarantee provided for the continuous and rapid reduction in PM2.5 concentrations in Changsha. While the PM2.5 concentrations in Changsha in 2019 and 2021 were slightly higher than the previous year's PM2.5 concentration, Changsha's overall PM2.5 concentration showed a rapidly decreasing trend. Between 2013 and 2021, the PM2.5 concentration in Changsha generally showed a trend in rapid reduction and was reduced by half.

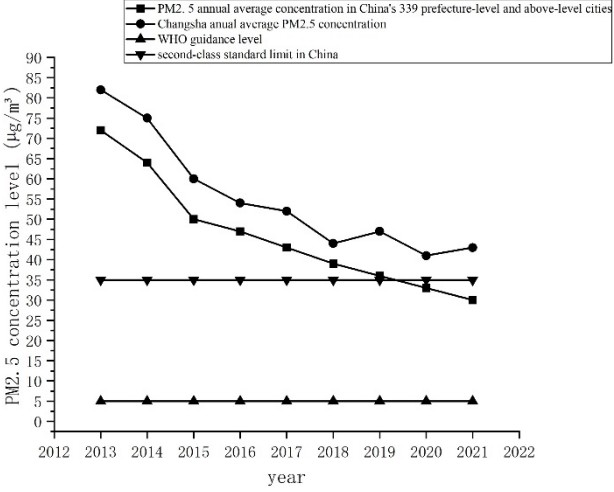

**Figure 2.** Annual average PM2.5 concentration in China and Changsha from 2013 to 2021 [41–52].

The results achieved during this period were attributed to the effective prevention and control measures taken by the Changsha Municipal People's Government against various PM2.5 pollution sources [39]. However, during these 9 years, the annual average PM2.5 concentration in Changsha was still above the WHO guideline for PM2.5 and the Level-2 limit for the PM2.5 concentration in China.

Researchers first analyze the error of simulated values. This enables an assessment to be made as to whether the simulated value can represent the actual situation well and can be used for correlation analysis. For error analysis, the researchers measured the PM2.5 concentration at a fixed point in the study area using TSI 8530 every hour from 13:00 and 23:00 on 24 August 2022 (Table 7). Model 1 was built according to actual conditions. The

error analysis in this study was based on the analysis of the measured and simulated values of each hour at the fixed points. In the ENVI-met model, the coordinate of the test point selected by the researchers was (58, 50). The measured and simulated PM2.5 concentrations at this test point in each hour are shown in the table below. The correlation between the measured PM2.5 concentration and the simulated PM2.5 concentration, calculated by the Spearman correlation coefficient in SPSS, was 0.902, and the significance is less than 0.001. In addition, the researchers calculated that the average absolute error between the simulated values and the measured values over the 10 h was 1.3 $\mu g/m^3$. Therefore, the simulation data have high reliability and can represent the actual PM2.5 concentration well.

**Table 7.** Hourly measured values of PM2.5 concentration at measurement points from 13:00 to 23:00 on 24 August 2022.

| Date | Time | Measured PM2.5 Concentration ($\mu g/m^3$) | Simulated PM2.5 Concentration ($\mu g/m^3$) |
|---|---|---|---|
| 24 August 2022 | 13:00–14:00 | 31 | 32 |
| | 14:00–15:00 | 28 | 31 |
| | 15:00–16:00 | 26 | 28 |
| | 16:00–17:00 | 25 | 25 |
| | 17:00–18:00 | 26 | 26 |
| | 18:00–19:00 | 29 | 27 |
| | 19:00–20:00 | 33 | 30 |
| | 20:00–21:00 | 35 | 34 |
| | 21:00–22:00 | 36 | 36 |
| | 22:00–23:00 | 37 | 36 |

Based on the five different models, the researchers simulated the distribution of PM2.5 pollution from 13:00 to 14:00 on 24 August 2022. Since the temperature, humidity, wind speed, wind direction, traffic volume, and PM2.5 pollution concentration in the surrounding that we collected during this period were fixed, the only difference between the five models is the height of the trees on both sides. Figure 3 below shows the concentration distribution of PM2.5 from 13:00 to 14:00 for the five models.

Figure 3 shows that the roadway had the highest PM2.5 pollution concentration. As the distance between the area and the roadway increased, the PM2.5 concentration was rapidly reduced. Due to the influence in the 232.0° southwesterly wind, there were more PM2.5 pollution areas on the east of the roadway with a PM2.5 pollution concentration between 31 and 34 $\mu g/m^3$ than on the west of the roadway.

In all five images, all areas where the PM2.5 concentration is higher than 39 $\mu g/m^3$ occurred on the roadway. In comparison, we can see that Model 5 with no street trees on both sides has the smallest area of PM2.5 greater than 39 $\mu g/m^3$, which is also significantly smaller than the corresponding parts in the other four models. To some extent, this supports Guo et al.'s (2018) [17] findings that road greening increases PM2.5 pollution on roadways. In addition, the researchers found that when the street trees are 10 m tall, the area where the PM2.5 concentration is higher than 39 $\mu g/m^3$ is largest. In the study, street trees with a height of 15 m have the largest crown diameter and crown width. However, we have found that it is not that the larger the crown diameter and crown width of street trees, the larger the area of extremely high PM2.5 pollution level (greater than 39 $\mu g/m^3$) on the roadway.

To further compare the PM2.5 concentration on roadways and sidewalks under different tree heights, the researchers selected five fixed points on roadways and six fixed points on sidewalks (three fixed points each on the west and east of the roadways) to compare the PM2.5 concentration at the same observation point under different models. In each model (Models 1 to 5), the researchers found the coordinates of the corresponding observation

points and obtained the corresponding PM2.5 concentration. The concentration of PM2.5 at each point is shown in Table 8 below.

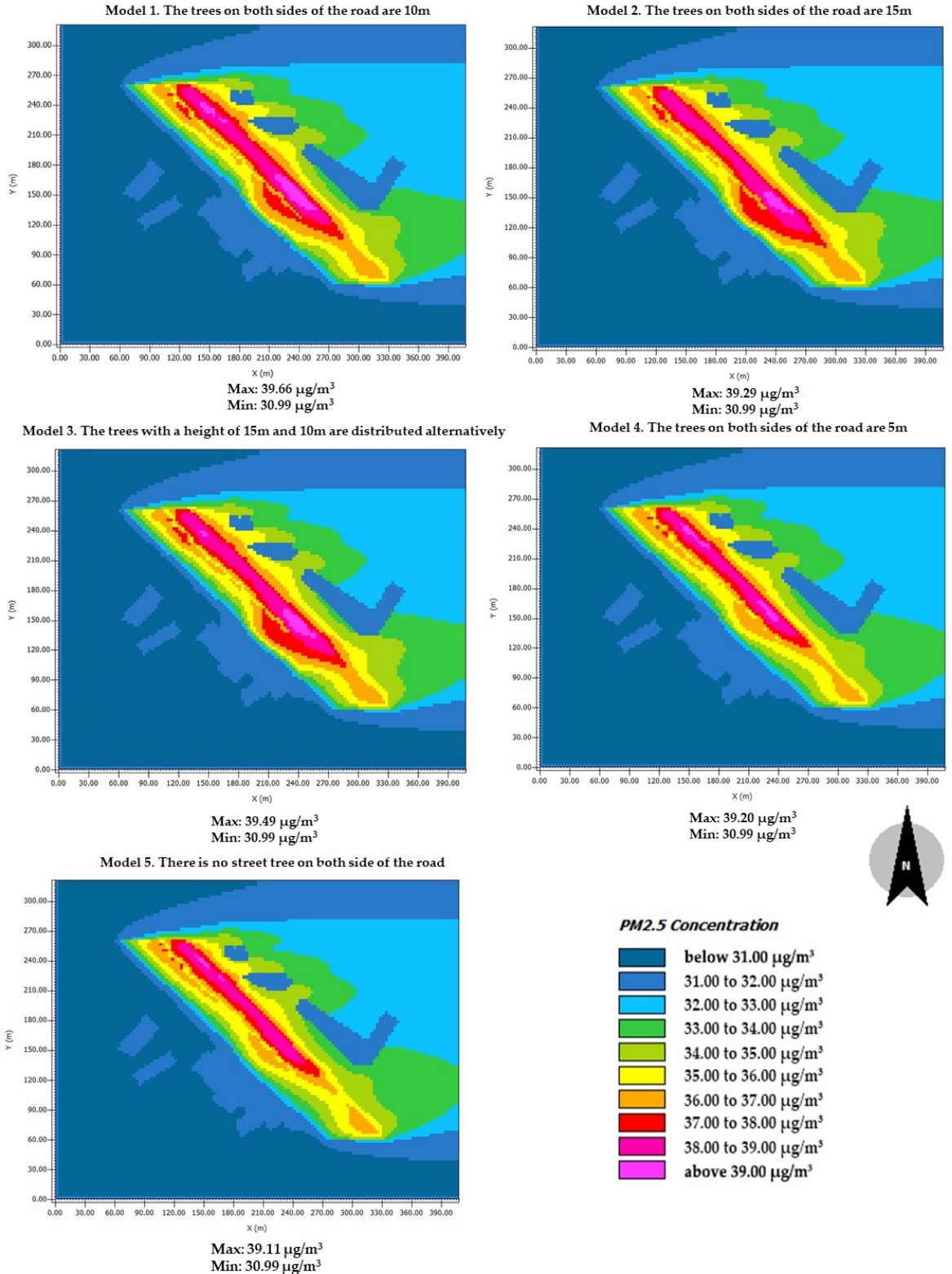

**Figure 3.** Simulation results of the distribution of the PM2.5 concentration with different street trees on both sides of the road.

**Table 8.** Statistics of the PM2.5 concentration at observation points in different road greening models.

| | | PM2.5 Concentration ($\mu g/m^3$) | | | | |
|---|---|---|---|---|---|---|
| | | Model 1. 10 m Trees on Both Sides of the Road | Model 2. 15 m Trees on Both Sides of the Road | Model 3. 15 and 10 m Trees Distributed Alternatively on Both Sides of the Road | Model 4. 5 m Trees on Both Sides of the Road | Model 5. No Tree on Both Sides of the Road |
| | Point 1 (38, 76) | 35.79 | 35.70 | 35.75 | 35.67 | 35.57 |
| | Point 2 (54, 62) | 35.06 | 35.01 | 35.06 | 35.01 | 34.99 |
| | Point 3 (74, 41) | 36.50 | 37.13 | 37.06 | 35.12 | 34.85 |
| | Point 4 (64, 63) | 38.32 | 38.11 | 38.18 | 38.20 | 38.12 |
| | Point 5 (97, 29) | 36.20 | 36.35 | 36.27 | 35.91 | 35.80 |
| Point number and coordinate | Point 6 (41, 66) | 31.05 | 31.04 | 31.05 | 31.06 | 31.06 |
| | Point 7 (66, 40) | 31.67 | 31.64 | 31.65 | 31.70 | 31.70 |
| | Point 8 (79, 30) | 31.91 | 31.89 | 31.89 | 31.94 | 31.95 |
| | Point 9 (74, 59) | 36.64 | 36.36 | 36.49 | 36.42 | 36.21 |
| | Point 10 (82, 51) | 36.74 | 36.58 | 36.73 | 35.93 | 35.72 |
| | Point 11 (90, 42) | 37.38 | 37.51 | 37.55 | 36.39 | 36.08 |

By comparing the PM2.5 concentration at five observation points (Observation Points 1 to 5) on the roadway, it can be seen that the PM2.5 concentration at each observation point on the roadway is lowest in models with no street trees on both sides. Through a comparison of Models 1 to 4, the researchers discovered that among the four models with street trees on both sides of different streets, the highest or lowest concentration of PM2.5 at the same observation point does not occur in any specific model. For example, from Observation Point 1, the concentration of PM2.5 is highest in Model 1 with 10 m street trees on both sides; however, from Observation Point 3, the concentration of PM2.5 is highest in Model 3 with 10 and 15 m street trees alternately distributed on both sides. Trees of different heights have different effects on the distribution of PM2.5 concentration on roadways. By comparing the PM2.5 concentration of observation points in the same coordinate (Observation Points 1 to 5) in different models, it can be seen that although most observation points have little difference in PM2.5 concentration, some show a significant difference. For example, in Observation Point 3, the PM2.5 concentration was 37.13 $\mu g/m^3$ in Model 2 and 34.85 $\mu g/m^3$ in Model 5.

According to the measured data by the researchers, the average wind direction during this period was 232.0°, which is a southwesterly wind. Observation Points 6 to 8 on the sidewalk on the west of the roadway were in the upwind direction, and Observation Points 9 to 11 on the sidewalk on the east of the roadway were in the downwind direction. In other words, when the wind blows through Observation Points 6 to 8, it has not yet passed the roadway polluted by traffic source PM2.5, but when the wind blows through Observation Points 9 to 11, it has passed the polluted roadway and carried a certain amount of PM2.5 pollution. By observing the PM2.5 concentration at Observation Points 6 to 8 on the west of the roadway among five different models, it can be seen that the PM2.5 concentration difference at the same observation point among five different models is very small, with only a difference between 0.01 and 0.06 $\mu g/m^3$, showing that different tree heights have little effect on PM2.5 concentration upwind. On the contrary, Observation Points 9 to 11, which were located downwind, show clear differences. Among different street tree height models, the difference in PM2.5 concentration in the same observation can be more than 1 $\mu g/m^3$. According to the standard control-diffusion equalization [26] mentioned above, it can be seen that environmental factors, such as temperature, humidity, wind speed, and wind direction affect the concentration distribution of PM2.5 in the air. In particular, this is due to the fact that plants can affect the surrounding microclimate. ENVI-met offers a variety of

tools to simulate and analyze the interactions between vegetation and microclimate [53]. In the model, the wind blows onto the roadway from the southwest, and more PM2.5 pollution spreads to the northeast due to the influence of the wind environment. Different plant heights have different effects on the surrounding microclimate, and different microclimate environments cause different diffusion patterns of PM2.5, which has a significant impact on the distribution of PM2.5 pollution. In addition, from the perspective of street trees, street trees can reduce PM2.5 pollution concentration through sedimentation, blocking, adsorption, and absorption. As evident in this study, street trees on the west of the roadway can influence the surrounding temperature, humidity, wind direction, wind speed, and other factors. Since the street trees on the west of the roadway were in the upwind direction, the street trees have a greater impact on the wind passing through them and blowing onto the roadway. The significant changes in the wind environment after passing through the street trees on the west of the roadway also resulted in uneven distribution of PM2.5 pollution concentration on the east of the roadway.

By comparing related data from Models 2 to 4, it was found in this study that there was no lowest PM2.5 pollution quality concentration among the models with alternately distributed street trees of different heights proposed by Karttunen et al. (2018) [14]. Different findings may be caused by the fact that this study considered only PM2.5, while Karttunen et al. (2018) [14], in their study, considered both PM2.5 and PM10. Furthermore, the wind direction setting in their study differed from this study. This study set the wind direction as 232.0° southwesterly wind according to the measured data, while they focused on the wind parallel and perpendicular to the street. They also clearly indicated in the study that depending on the scenario and direction of the wind relative to the boulevard, planting trees increases the mean PM10 by 4% to 123% and PM2.5 by 1% to 72%.

The crown widths of 15, 10, and 5 m camphor trees were 11, 7, and 3 m, respectively. In Model 4, the street trees had a height of 5 m and a crown width of 3 m. Compared with Models 1 to 3, in Model 4, there were more empty spaces between street trees, with a relatively small total volume of plants that could influence the microclimate and a significantly smaller total volume of plants that could sediment, block, sorb, and absorb PM2.5. From Observation Points 1 to 8 in Figure 3 and Table 2, it can be seen that the street trees with a crown width of 3 m in Model 4 and the street trees with a crown width of 7 and 11 m distributed alternately in Models 1, 2, and 3, respectively, had a small difference in terms of the impact on the PM2.5 concentration on the west sidewalk and roadway. Based on the data of each observation point, in general, the PM2.5 concentration in Observation Points 6 to 8 on the west side of the road in Models 1 to 4 is not very different. According to the comparison between the PM2.5 concentration corresponding to Observation Points 1 to 5 on the roadway in Models 1 to 4, when the height of the roadside trees on both sides is 5 m and the crown width is 3 m (Model 4), the PM2.5 pollution concentration on the roadway is generally the lowest. When the height of the roadside trees on both sides is above or equal to 10 m and the crown width is above or equal to 7 m, the distance between those trees is shorter. An increase in their crown width would not increase the concentration of PM2.5 on the roadway accordingly. Meanwhile, from Observation Points 9 to 11 on the east sidewalk, it can be found that when the crown width of street trees was 3 m, the observation points on the east sidewalk had a lower PM2.5 concentration than the observation points at the corresponding coordinate points in Models 1 to 3. In Models 1 to 3, the crown width of all street trees was greater than or equal to 7 m, and the PM2.5 concentration levels at the observation points with the same coordinates had little difference, and there was no discernible pattern to the differences.

In the models without roadside trees, PM2.5 pollution from motor vehicles can easily diffuse to the surrounding environment, leading to a lower PM2.5 concentration around the road than the models with roadside trees. However, if plants do not reduce PM2.5 pollution from motor vehicles in time, it is unfavorable to the control of PM2.5 pollution from road traffic sources. Numerous tall roadside trees with large crown width have a large number of leaves that can reduce PM2.5 pollution, while at the same time, this may block the wind

environment and cause PM2.5 pollution to accumulate on roadways and sidewalks. The street canyon width, the features of the surrounding buildings, the roadway traffic volume, and the microclimate conditions are varied for different roads. The diffusion of PM2.5 emitted by motor vehicles in different environments is also varied. The researchers believe that how to adjust the height of the roadside trees to bring into full play the role of plants for PM2.5 pollution reduction and reduce the concentration of PM2.5 on the sidewalk as much as possible need to be determined according to the specific circumstances. The researchers found that the wind environment has a very clear effect on PM2.5 pollution, the reasonable ventilation makes PM2.5 pollution diffusion and reduces PM2.5 pollution in a high value area.

## 4. Conclusions

In this study, the researchers select a section of Shaoshan South Road, a representative two-way six-lane road in the central urban area of Changsha City, a capital city in Central China, as the object. The roadside trees in this section are mainly camphor trees. Camphor trees are also the major roadside trees in the central urban area of Changsha City. Through a constructing model according to the actual situation (the height of roadside trees on both sides is about 10 m), the researchers constructed another four contrast scenario models including no roadside trees on both sides, 15 m high roadside trees on both sides, 5 m high roadside trees on both sides, and alternate distribution of 15 and 10 m high roadside trees on both sides to explore the impact of height of roadside trees on road PM2.5 concentration. The research has found that roadside trees can lead to the increase in PM2.5 concentration in the roadway and sidewalk, and in the model without roadside trees on both sides, PM2.5 concentration of each point in the area is relatively low. When roadside trees on both sides are 5 m high, the crown breadth is about 3 m, thus there is a larger space among the roadside trees. In this case, the PM2.5 concentration of the sidewalk on the east side of the roadway and in the downwind direction is lower than the scenario model where the height of roadside trees is 10 m and the crown breadth is up to 7 m and above. However, when the height and crown breadth of roadside trees are respectively greater than or equal to 10 and 7 m, the increase in height and crown breadth of roadside trees fails to cause the regular increase or decrease in PM2.5 concentration around them. Although plants can reduce the concentration of PM2.5 in the air through settlement, retardation, adherence, and absorption, plants can make PM2.5 pollution gather in the narrow space of street canyons due to the relatively narrow space there, thus increasing the concentration of PM2.5. Reducing air pollution is an important function of plants. When plants are dense in street canyons, the alternate distribution of plants with different heights forms height difference among the plants, causing the effect that air circulation does not necessarily reduce PM2.5 concentration. From the perspective of good ventilation, the height of roadside trees should not be too high and the crown breadth should not be too big. When selecting plants, it is necessary to fully consider their impact on the wind environment in order to ensure good wind passage and facilitate the diffusion of PM2.5 pollution. Ensuring good ventilation is only conducive to the diffusion of PM2.5 pollution, never substantially reducing PM2.5 pollution. Only when the crown diameter and breadth of roadside trees are in a certain scale, plants can better play their role in settling, retarding, adhering, and absorbing PM2.5 pollution from traffic. Therefore, it is imperative to fully consider the road conditions, and reasonably select the plant's height and crown breadth to achieve a good balance between "ventilation" and "reduction".

**Author Contributions:** Conceptualization, J.L. and B.Z.; methodology, J.L. and Y.X.; software, J.L. and Y.X.; validation, J.L., J.F. and B.Z.; formal analysis, J.L.; investigation, J.L. and J.F.; resources, J.L. and J.F.; data curation, J.L. and B.Z.; writing—original draft preparation, J.L.; writing—review and editing, J.L. and J.F.; visualization, J.L. and Y.X.; supervision, B.Z.; project administration, J.L. All authors have read and agreed to the published version of the manuscript.

**Funding:** This research received no external funding.

**Institutional Review Board Statement:** Not applicable.

**Informed Consent Statement:** Not applicable.

**Data Availability Statement:** Not applicable.

**Conflicts of Interest:** The authors declare no conflict of interest.

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
