# Peer review of "The Impact of Street Tree Height on PM2.5 Concentration in Street Canyons: A Simulation Study"

_sustainability, doi:10.3390/su141912378_

Round 1
Reviewer 1 Report
Dear Authors,
The paper you proposed to be published in this journal proves a hard work and a deep knowledge of the researched aspects.
Here are some observations and recommendations for the further improvement of the paper:
Parts of the manuscript need to be completed (such as, for example, type of the article, keywords and Back Matter)
In the Introduction section:
-line 53: please consider "Figure 1" instead of "Table 1"
-lines 58 and 62: is it United Nations or World Health Organization?
-please add the test hypothesis of the study.
In the Materials and methods section:
-the first paragraph of this section should be moved to the "Introduction" section
-line 202: you probably meant "Figure 2". The provider of the image (the source) should be specified
-line 220: please consider "Table 1" instead of "Table 2"
-line 224: please introduce the references numbers in parenthesis.
In the Results and Discussion section:
-line 246: you mentioned "five different heights of trees", but only three heights were considered in the study: 15 m, 5 m, and 2 m; please clarify
-in Figure 3, the writing above the second image seems not to be fully visible
-line 257: are you sure it is "minimum" and not "maximum" 33.78 μg/m3 ?
-please express the values as "μg", instead of "ug", including in Table 2
-please also refer to the results of other studies in the literature.
The References section: please follow the style recommended by the journal in the "Instructions for authors" section.
Author Response
Thank you for giving us the chance to revise the article. A reviewer asked us to provide error analysis. We can’t analyze error based on the original data of June 27, 2022 without the collection of measured PM2.5 concentration. Having re-selected the situation of this study area on August 24, 2022, we carried out the actual measurement and simulation study to collect the data related to microclimate. Therefore, we finished the error analysis to get the result that the simulation result can represent the actual situation well. Since we did the experiment again, we have basically reworked this article from the experiment to the analysis.
By the way, I deem that the paper document you received this time is a document in which the contents submitted for the first time and those submitted after revision in this article are arranged together in a rather disorderly manner according to my previous contribution in sustainability. This is not because of the mistakes in our typesetting, but the revision symbol required by the journal to be used in microsoft for revising the original text. We submitted the paper to the system which deleted all the deletion symbols and the result was like that. I suggest that you judge what we have deleted and added by comparing the version we submitted for the first time with the version submitted this time. At the end of the answer to this question, we attached our abstract to help you understand the general content of the revised article.
We did spend more than 40 days to revise this article with effort. I hope you can give us another chance to revise it if you think it is possible to publish the paper.
Point 1: Parts of the manuscript need to be completed (such as, for example, type of the article, keywords and Back Matter)
Response 1: We're terribly sorry. We chose article as the paper type. There may have been a slight error in the system when generating the document. Anyway, we've corrected all the details. We have used ‘article’ as the paper type. We have added relevant key words.
Point 2: line 53: please consider "Figure 1" instead of "Table 1"
Response 2: We are sorry for the negligence. We have changed from "Table 1" to "Figure 1".
Point 3: -lines 58 and 62: is it United Nations or World Health Organization?
Response 3: We are sorry, it should be ‘World Health Organization’. When we wrote the article, we thought it can be replaced by “United Nations”, because World Health Organization is a part of United Nations. Thank you for your reminding, we have realized, in this context, we cannot use ‘United Nations’. In line 58 and 62, we have changed from ‘United Nations’ to “World Health Organization”. In addition, another reviewer suggested ‘Lines 82-124, this part should also be significantly resized and moved in the Discussions section.’ We have already resized this part and moved it in the Discussions section.
Point 4: -please add the test hypothesis of the study.
Response 4: Does the test hypothesis means error analysis? We have provided error analysis now, we first recorded the hourly measured average PM2.5 concentration at one observation point within the study scope for 10 hours with the test method. We used TSI8530 as the instrument with an error of ±0.002 mg/m3. After completing the hourly simulation of the study area, we extracted the corresponding hourly simulation value of this observation point. We calculated the mean absolute error between the simulated value and the measured value within 10 hours, which was 1.3μg/m3. We calculated the correlation between the measured PM2.5 concentration and the simulated PM2.5 concentration through the Spearman correlation analysis, which was 0.902. The significance was less than 0.001. We believe that the related results can show that the study has high reliability.
Point 5: In the Materials and methods section:
-the first paragraph of this section should be moved to the "Introduction" section
Response 5: We have moved the first paragraph to the ‘introduction’ section. In addition, another reviewer suggested ‘this paragraph should be drastically reduced’. We have already drastically reduced this paragraph. You can see the simplified information in the last paragraph of the ‘introduction’ section.
Point 6: -line 202: you probably meant "Figure 2". The provider of the image (the source) should be specified
Response 6: We are sorry. The figure number in line 202 should be figure 2. Because another reviewer suggested us replace this figure with a three-dimensional representation of the model. We used sketch up to build a three-dimensional model of the research area and use a new figure to replace this one. Anyway, the figure number in the paragraph consistent with the figure number in the figure title now.
Point 7: -line 220: please consider "Table 1" instead of "Table 2"
Response 7: we are sorry. The table number should be table 1. We have corrected the mistake.
Point 8: -line 224: please introduce the references numbers in parenthesis.
Response 8: We have introduced the references numbers in parenthesis.
In the Results and Discussion section:
Point 9: -line 246: you mentioned "five different heights of trees", but only three heights were considered in the study: 15 m, 5 m, and 2 m; please clarify
Response 9: Sorry, this is because we made a small mistake in the process of translating the article from Chinese to English. What we are trying to say are the five situations that arise from different heights of street trees, which respectively are: 1. No street trees on both sides 2. 5-m high street trees on both sides 3. 10-m high street trees on both sides 4. 15-m high street trees on both sides 5. Alternate distribution of 15-m and 10-m high street trees on both sides
Point 10: -in Figure 3, the writing above the second image seems not to be fully visible
Response 10: We are sorry. Since we have carried out the simulation again and generate new pictures, we have deleted this picture. We have carefully checked and ensured that our newly generated image does not have this problem. You may also see this picture. It's not that we didn't delete this picture, but that the submission system of MDPI will mix the revised version with the previous version to facilitate reviewers to compare which parts have been revised. We suggest you compare the version you downloaded during the first round of review with the document you see now. In this way, you can easily know what we have modified.
Point 11: -line 257: are you sure it is "minimum" and not "maximum" 33.78 μg/m3 ?
Response 11:It should be ‘maximum’. We are sorry for the negligence. We will try our best to avoid similar mistake happen again.
Point 12: please express the values as "μg", instead of "ug", including in Table 2
Response 12:We are sorry, we have corrected the mistake in the whole article.
Point 13: please also refer to the results of other studies in the literature.
Response 13:We have referred to the results of other studies in the literature.
Point 14: The References section: please follow the style recommended by the journal in the "Instructions for authors" section.
Response 14:I'm really sorry, I've been trying my best to revise my article for more than 40 days. But the time is very tight, and now there is only one hour left, it's not possible for me to set the type completely according to the MDPI citation format within such a short time. If you give us another chance for revision again, I will put the reference format completely right.
Abstract of the article
With the rapid development of cities and the rapid increase in automobile ownership, traffic has become one of the main sources of PM2.5 pollution, which can be reduced by road greening through sedimentation, blocking, adhesion, and absorption. Using the method of combining field monitroing and ENVI-met simulation, the influence of the tree height on the PM2.5 concentration on both sides of the city streets was discussed. The influence of tree height on PM2.5 under five conditions was analyzed, including 10-meter-tall trees (i), 15-meter-tall trees (ii), alternating distribution of 15-meter-tall and 10 -meter-tall trees (iii), 5-meter-tall trees (IV), no trees on either side of the road (V). The results show that: Roadside trees can increase the concentration of PM2.5 in the narrow space of street canyons. However, without roadside trees, PM2.5 from traffic sources is not reduced in time, it is more easily spread to the distance. The height of the roadside trees is 5 meters, and their crown widths are smaller than those of other trees, and there is a relatively wide space between those trees. Compared with the higher roadside tree models with larger crown widths, the concentration of PM2.5 on the roadway and the downwind sidewalk is relatively low. In the three models (i- iii) with tree height above or equal to 10 meters, the PM2.5 concentration around the trees do not show regular change with the change of tree height. Due to the tree height of 10 meters and 15 meters, the crown width is larger, and the alternate distribution of tree height of 15 meters and 10 meters fails to make the PM2.5 concentration in the models lower than that in the models with tree height of 15 meters or 10 meters. The reasonable height of roadside trees in street canyons helps improve the wind circulation to promote the diffusion of PM2.5 pollution. There’s no optimal height of roadside trees for PM2.5 pollution in street canyons, so it is necessary to select the height reasonably according to the specific situation.
Reviewer 2 Report
Journal: MDPI, Sustainability
Review of Manuscript #1845816
Title: “The impact of street tree height on PM2.5 concentration in street canyons: A simulation study”
By Liu et al.
General comment.
The present manuscript proposes the application of a predictive software model for estimating the mitigation of PM2.5 concentrations in an urban area, as a function of the distribution and height of trees in a simulated context. Although the topic is potentially interesting, I believe that the manuscript currently has several problems: 1) the general organization of the text should be completely revised, as some parts would need to be drastically summarized or eliminated, while others should be moved to different sections of the manuscript; 2) the application of the predictive model seems to have some biases that should necessarily be solved, such as the fact that the tree species is not considered; 3) there is a complete lack of a description of the estimate of the error produced by the predictive method, of the confidence limits of the values ​​generated and, more generally, a robust validation of the method itself with a description of its reproducibility, reliability, accuracy and limits; 4) the study should consider several urban scenarios in comparison (for example at least 3), characterized by different levels of atmospheric contamination. In this sense, I believe that this manuscript needs a drastic revision before being considered for publication. Specific comments are included below.
Specific comments.
Whole manuscript, in general: most of the cited literature in the present manuscript refers to studies from China, obviously I do not question the relevance of the cited publications, but it would be advisable for the authors to increase the panorama of the used sources by including more international literature, contributing also to increase the level of international relevance of their manuscript, which at the moment appears to have a regional interest.
Introduction in general: this part appears too extended and should be drastically reduced; some parts may be moved elsewhere, while other parts should be summarized. Specifically: lines 47-63, these topics should be moved to the Results and Discussion section (see also the next comment). Lines 82-124, this part should also be significantly resized and moved in the Discussions section.
Introduction: Figure 1 and related data should be moved into Results and Discussion section; authors should also include a description into Mat&Met section related to the source and elaborations of such data.
Introduction: conventionally the genus and species are written in full only the first time these are named, subsequently the genus should be indicated only with the first letter followed by a dot, i.e .: Podocarpus macrophyllus later it should be indicated as P. macrophyllus. Authors should apply this modification to the entire manuscript for each indicated species.
Material and Methods in general: also in this case the text appears too extensive and contains information that goes beyond the context of the paragraph dedicated to materials and methods. For example, the geographical description (Lines 157-186) of the selected site should be drastically reduced, or even better eliminated: authors could include this information in a supplementary document to be attached to the manuscript.
Materials and Methods: Figure 2 does not seem to provide useful elements to the context, this should be removed or replaced with a more suitable figure, for example a three-dimensional representation of the model, rather than an aerial photo.
Lines 189-199: considering that the present study is based on the application of a mathematical model and a microclimatic simulation, applied by a software, the authors should provide more robust information related to the validation of the method, including the estimation of the predictive error, of the confidence limits, as well as describing the applications limits of the model and any disturbing or interference factors.
Lines 200-222: considering that the present study aims to evaluate the interaction of the distribution and height of trees in an urban context in the mitigation of atmospheric levels of PM2.5, I am surprised that the proposed simulation does not consider the trees species any way: I think that the specific characteristics of plants can drastically influence phenomena such as sedimentation, blocking, adhesion, and absorption (cited by the same authors in the abstract) and therefore modulate the mitigation of PM2.5 levels. In this sense, such specific characteristics play as important a role as the distribution and height of the plants and should be considered. Authors should provide comment.
Table 1 and related data should be moved into Results and Discussion section.
Results and Discussion in general: in the description of the results authors should include the estimation of the error and of the confidence limits associated with the proposed simulations.
Figure 3: the text included in the present figure appears very small and sometimes grainy, of poor graphic quality, authors should provide an image with a better graphic quality level and increase the size of the text.
Table 2 and related data: authors should provide a confidence interval for all the estimated levels of PM2.5, otherwise the data is not statistically relevant. In addition, as already commented, it is essential that the authors provide data relating to the estimation of the error of the model and the prediction, without which this manuscript risks having a poor level of reliability and accuracy.
Author Response
Thank you for giving us the chance to revise the article. You asked us to provide error analysis. We can’t analyze error based on the original data of June 27, 2022 without the collection of measured PM2.5 concentration. Having re-selected the situation of this study area on August 24, 2022, we carried out the actual measurement and simulation study to collect the data related to microclimate. Therefore, we finished the error analysis to get the result that the simulation result can represent the actual situation well. Since we did the experiment again, we have basically reworked this article from the experiment to the analysis.
By the way, I deem that the paper document you received this time is a document in which the contents submitted for the first time and those submitted after revision in this article are arranged together in a rather disorderly manner according to my previous contribution in sustainability. This is not because of the mistakes in our typesetting, but the revision symbol required by the journal to be used in microsoft for revising the original text. We submitted the paper to the system which deleted all the deletion symbols and the result was like that. I suggest that you judge what we have deleted and added by comparing the version we submitted for the first time with the version submitted this time. At the end of the answer to this question, we attached our abstract to help you understand the general content of the revised article.
We did spend more than 40 days to revise this article with effort. I hope you can give us another chance to revise it if you think it is possible to publish the paper.
Point 1:
Whole manuscript, in general: most of the cited literature in the present manuscript refers to studies from China, obviously I do not question the relevance of the cited publications, but it would be advisable for the authors to increase the panorama of the used sources by including more international literature, contributing also to increase the level of international relevance of their manuscript, which at the moment appears to have a regional interest.
Response 1: We have added many international literatures.
Point 2: Introduction in general: this part appears too extended and should be drastically reduced; some parts may be moved elsewhere, while other parts should be summarized. Specifically: lines 47-63, these topics should be moved to the Results and Discussion section (see also the next comment). Lines 82-124, this part should also be significantly resized and moved in the Discussions section.
Response 2: We have moved lines 47-63 from introduction to discussion. We have significantly resized and moved in the Discussions section.
Point 3: Introduction: Figure 1 and related data should be moved into Results and Discussion section;
Response 3: Figure 1 and related data have been moved into Results and Discussion section
Point 4: Introduction: conventionally the genus and species are written in full only the first time these are named, subsequently the genus should be indicated only with the first letter followed by a dot, i.e .: Podocarpus macrophyllus later it should be indicated as P. macrophyllus. Authors should apply this modification to the entire manuscript for each indicated species.
Response 4: Thanks quite a lot for your valuable suggestion. The genus and species are written in full the first time these are named, subsequently the genus are indicated with the first letter followed by a dot.
Point 5: Material and Methods in general: also in this case the text appears too extensive and contains information that goes beyond the context of the paragraph dedicated to materials and methods. For example, the geographical description (Lines 157-186) of the selected site should be drastically reduced, or even better eliminated: authors could include this information in a supplementary document to be attached to the manuscript.
Response 5: We have drastically reduced the geographical description (Lines 157-186).
Point 6: Materials and Methods: Figure 2 does not seem to provide useful elements to the context, this should be removed or replaced with a more suitable figure, for example a three-dimensional representation of the model, rather than an aerial photo.
Response 6: We have deleted the aerial photo. We have used a three-dimensional representation of the model to replace the aerial photo.
Point 7: Lines 189-199: considering that the present study is based on the application of a mathematical model and a microclimatic simulation, applied by a software, the authors should provide more robust information related to the validation of the method, including the estimation of the predictive error, of the confidence limits, as well as describing the applications limits of the model and any disturbing or interference factors.
Response 7: We have provided estimation of the predictive error. We first recorded the hourly measured average PM2.5 concentration at one observation point within the study scope for 10 hours with the test method. We used TSI8530 as the instrument with an error of ±0.002 mg/m3. After completing the hourly simulation of the study area, we extracted the corresponding hourly simulation value of this observation point. We calculated the mean absolute error between the simulated value and the measured value within 10 hours, which was 1.3μg/m3. We calculated the correlation between the measured PM2.5 concentration and the simulated PM2.5 concentration through the Spearman correlation analysis, which was 0.902. The significance was less than 0.001. We believe that the related results can show that the study has high reliability.We have not found any disturbing or interference factor at the moment.
Point 8: Lines 200-222: considering that the present study aims to evaluate the interaction of the distribution and height of trees in an urban context in the mitigation of atmospheric levels of PM2.5, I am surprised that the proposed simulation does not consider the trees species any way: I think that the specific characteristics of plants can drastically influence phenomena such as sedimentation, blocking, adhesion, and absorption (cited by the same authors in the abstract) and therefore modulate the mitigation of PM2.5 levels. In this sense, such specific characteristics play as important a role as the distribution and height of the plants and should be considered. Authors should provide comment.
Response 8: We are sorry. We have specified the tree species information their crown width and leaf area density in the table.
Point 9: Table 1 and related data should be moved into Results and Discussion section.
Response 9: Figure 1 and related data have been moved into Results and Discussion section.
Point 10: Results and Discussion in general: in the description of the results authors should include the estimation of the error and of the confidence limits associated with the proposed simulations.
Response 10:
ENVI-met simulates the distribution of PM2.5 pollution in the environment relying on computational fluid dynamics, standard convection diffusion equation, etc. They have a module specially used to simulate air pollutants (including the simulation of PM2.5). We need to first build the model (build the model of buildings and roads with information such as length, width, height, and shape), and then add materials, vegetation, and others to the model. After that, we need to input microclimate information (temperature, humidity, wind direction, wind speed, etc.) and information related to PM2.5 from traffic sources in the study area, such as the number of lanes and the number of motor vehicles. We cannot directly obtain the predicted data of all the points and all the periods like predicting with a lot of programming software such as pyhton. We selected to illustrate that the research error was relatively small through hourly measured data of 10 hours and simulation values of PM2.5 of corresponding points.
Besides, we would like to give supplementary instruction. Software developers have verified that the results of the simulation of PM2.5 are generally reliable when they develop the software. Some researchers have used this software to make the simulation analysis and publish papers. The reliability of simulation results has also been recognized by some people.
We first recorded the hourly measured average PM2.5 concentration at one observation point within the study scope for 10 hours with the test method. We used TSI8530 as the instrument with an error of ±0.002 mg/m3. After completing the hourly simulation of the study area, we extracted the corresponding hourly simulation value of this observation point. We calculated the mean absolute error between the simulated value and the measured value within 10 hours, which was 1.3μg/m3. We calculated the correlation between the measured PM2.5 concentration and the simulated PM2.5 concentration through the Spearman correlation analysis, which was 0.902. The significance was less than 0.001. We believe that the related results can show that the study has high reliability.
Point 11: Figure 3: the text included in the present figure appears very small and sometimes grainy, of poor graphic quality, authors should provide an image with a better graphic quality level and increase the size of the text.
Response 11: Since we have carried out the simulation again and generate new pictures, we have deleted this picture. We have carefully checked and ensured that our newly generated image does not have this problem. You may also see this picture. It's not that we didn't delete this picture, but that the submission system of MDPI will mix the revised version with the previous version to facilitate reviewers to compare which parts have been revised. We suggest you compare the version you downloaded during the first round of review with the document you see now. In this way, you can easily know what we have modified.
Point 12: Table 2 and related data: authors should provide a confidence interval for all the estimated levels of PM2.5, otherwise the data is not statistically relevant. In addition, as already commented, it is essential that the authors provide data relating to the estimation of the error of the model and the prediction, without which this manuscript risks having a poor level of reliability and accuracy.
Response 12:
Because we didn’t expect that you would ask us to make the error analysis, we didn’t measure the corresponding actual data of the simulation period at that time. For the error analysis, we did another round of study (the recollected measured data included temperature, humidity, wind direction, wind speed, the PM2.5 concentration, etc.) and made the error analysis according to the measured data and simulation data.
ENVI-met simulates the distribution of PM2.5 pollution in the environment by using computational fluid dynamics and standard convection diffusion equation. ENVI-met software does not predict the PM2.5 concentration with the traditional machine learning method. The difference from machine learning is that the software calculates the PM2.5 concentration at the corresponding point by related formula after a spatial model is built and data such as air temperature, humidity, and wind speed are input in ENVI-met and the value of the same observation point simulated in the same model is always the same fixed value. We don’t know how to provide the confidence interval or degree of confidence. We believe that the mean absolute error of 10 hours can also show the error in the experiment. We calculated the mean absolute error and Spearman correlation coefficient.
We would like to illustrate the error of simulation data with the Spearman correlation coefficient. We first recorded the hourly measured average PM2.5 concentration at one observation point within the study scope for 10 hours with the test method. We used TSI8530 as the instrument with an error of ±0.002 mg/m3. After completing the hourly simulation of the study area, we extracted the corresponding hourly simulation value of this observation point. We calculated the mean absolute error between the simulated value and the measured value within 10 hours, which was 1.3μg/m3. We calculated the correlation between the measured PM2.5 concentration and the simulated PM2.5 concentration through the Spearman correlation analysis, which was 0.902. The significance was less than 0.001. We believe that the related results can show that the study has high reliability.
Abstract of the article
With the rapid development of cities and the rapid increase in automobile ownership, traffic has become one of the main sources of PM2.5 pollution, which can be reduced by road greening through sedimentation, blocking, adhesion, and absorption. Using the method of combining field monitroing and ENVI-met simulation, the influence of the tree height on the PM2.5 concentration on both sides of the city streets was discussed. The influence of tree height on PM2.5 under five conditions was analyzed, including 10-meter-tall trees (i), 15-meter-tall trees (ii), alternating distribution of 15-meter-tall and 10 -meter-tall trees (iii), 5-meter-tall trees (IV), no trees on either side of the road (V). The results show that: Roadside trees can increase the concentration of PM2.5 in the narrow space of street canyons. However, without roadside trees, PM2.5 from traffic sources is not reduced in time, it is more easily spread to the distance. The height of the roadside trees is 5 meters, and their crown widths are smaller than those of other trees, and there is a relatively wide space between those trees. Compared with the higher roadside tree models with larger crown widths, the concentration of PM2.5 on the roadway and the downwind sidewalk is relatively low. In the three models (i- iii) with tree height above or equal to 10 meters, the PM2.5 concentration around the trees do not show regular change with the change of tree height. Due to the tree height of 10 meters and 15 meters, the crown width is larger, and the alternate distribution of tree height of 15 meters and 10 meters fails to make the PM2.5 concentration in the models lower than that in the models with tree height of 15 meters or 10 meters. The reasonable height of roadside trees in street canyons helps improve the wind circulation to promote the diffusion of PM2.5 pollution. There’s no optimal height of roadside trees for PM2.5 pollution in street canyons, so it is necessary to select the height reasonably according to the specific situation.
Reviewer 3 Report
This manuscript requires significant changes before acceptance. Please find my comments below:
*In the abstract, points 1, 2, and 3 in lines 16-20 present the same information, please avoid repetition. I could not see the main results (numbers) in the abstract.
*keywords are missing.
*Line 39: add abbreviation of World Health Organization (WHO). Lines 41 and 43: replace World Health Organization by WHO.
*In the introduction: create a link between your research and the UN SDGs.
*Line 53: Replace “Table 1” by “Figure 1”.
*Line 56: is it 25 mg/m3 or 35 mg/m3?
*Lines 58-59: you mentioned that “During these seven years, the PM2.5 concentration in Changsha generally showed a trend of rapid reduction and was reduced by half.”, please provide an explanation for that.
* Need reference for data in Figure 1 & Table 1.
*Lines 77-78: delete this statement, the same meaning as statement in lines 74-77.
*Need reference for the statement in lines 50-51 & lines 82-84 & lines 88-97 & lines 134-140 & lines 223-224.
*Lines 113-115: so what? What is the main findings and conclusions of Hu (2019)?
*Line 144: which study? Need reference.
*Lines 151-155: grammatical errors, please rephrase.
*Please divide the Materials and Methods section to sub-sections (2.1. Study area, 2.2. simulation software).
*Lines 167-186: more suitable to be included to the introduction section.
*You did not refer to Figure 2 in the text.
*Figure 2, need to include more data: north direction, scale, grid line, etc.
*Line 191: Bruse et al., could not find it in the references list.
*Line 202: do you mean Figure 2? *Line 220: do you mean Table 1? *Line 245: do you mean Figure 2?
*Please add a table at the end of Materials and Methods section that include list of inputs data you entered to the model for each scenario.
*Lines 254-258: be consistent when using units ug and μm?
*Lines 273-275: need evidence, reference.
*Need to see some results in the conclusion section. I have notice repetition for some information in the conclusion section.
Author Response
Thank you for giving us the chance to revise the article. A reviewer asked us to provide error analysis. We can’t analyze error based on the original data of June 27, 2022 without the collection of measured PM2.5 concentration. Having re-selected the situation of this study area on August 24, 2022, we carried out the actual measurement and simulation study to collect the data related to microclimate. Therefore, we finished the error analysis to get the result that the simulation result can represent the actual situation well. Since we did the experiment again, we have basically reworked this article from the experiment to the analysis.
By the way, I deem that the paper document you received this time is a document in which the contents submitted for the first time and those submitted after revision in this article are arranged together in a rather disorderly manner according to my previous contribution in sustainability. This is not because of the mistakes in our typesetting, but the revision symbol required by the journal to be used in microsoft for revising the original text. We submitted the paper to the system which deleted all the deletion symbols and the result was like that. I suggest that you judge what we have deleted and added by comparing the version we submitted for the first time with the version submitted this time. At the end of the answer to this question, we attached our abstract to help you understand the general content of the revised article.
We did spend more than 40 days to revise this article with effort. I hope you can give us another chance to revise it if you think it is possible to publish the paper.
Point 1: This manuscript requires significant changes before acceptance.
Response 1: Thanks for your acknowledgement. We have tried our best to revised the article. We hope you think the revised version is good.
Point 2: *In the abstract, points 1, 2, and 3 in lines 16-20 present the same information, please avoid repetition. I could not see the main results (numbers) in the abstract.
Response 2: We are sorry. I have rewritten the abstract to solve the issue. Since we did the experiment again, we have basically reworked this article from the experiment to the analysis including the abstract.
Point 3: *keywords are missing.
Response 3: We have added relevant key words.
Point 4: *Line 39: add abbreviation of World Health Organization (WHO). Lines 41 and 43: replace World Health Organization by WHO.
Response 4: Thanks for your advice. We have added abbreviation of World Health Organization. We have replaced World Health Organization by WHO.
Point 5: *In the introduction: create a link between your research and the UN SDGs.
Response 5: We have created a link between our research and the UN SDGs in the introduction. It can be seen from the first eight lines of the introduction ‘’. In addition, the first two line of the Second paragraph of the introduction also shows the link between our research and the UN SDGs.
Point 6: *Line 53: Replace “Table 1” by “Figure 1”.
Response 6: We am sorry for the negligence. We have replaced “Table 1” by “Figure 1” in Line 53. By the way, we have moved the figure from introduction part to discussion part to meet the requirement of another reviewer.
Point 7: *Line 56: is it 25 mg/m3 or 35 mg/m3?
Response 7: We are sorry, it should be 25μg/m3 and 35 μg/m3. We have corrected all relevant errors in the article.
Point 8: *Lines 58-59: you mentioned that “During these seven years, the PM2.5 concentration in Changsha generally showed a trend of rapid reduction and was reduced by half.”, please provide an explanation for that.
Response 8: We have provided the explanation in the article. In addition, another reviewer suggested us that ‘Lines 82-124 should be resized and moved in the Discussions section’. we have moved line 47-63 from introduction part to the first paragraph of the discussion part to meet the requirement of another reviewer. You can find the explanation in the first paragraph of the discussion section.
The explanation is ‘To respond to the state’s call for PM2.5 pollution abatement and meet the national goal of reaching a PM2.5 pollution of between 15 and 25 μg/m3, which is the lower limit, among most developed countries by 2035. The Changsha Municipal People’s Government proposed in the “13th Five-Year” Ecological Construction and Environmental Protection Plan of Changsha City (2016-2020) that Changsha’s annual average PM2.5 concentration in 2020 should be reduced by 18% of that in 2015. Even if, in 2019, the PM2.5 concentration in Changsha was slightly higher than the previous years. From 2012 to 2020, the PM2.5 concentration in Changsha generally showed a trend of rapid reduction and was reduced by half. The results achieved during this period were attributed to the effective prevention and control measures taken by the Changsha Municipal People’s Government against various PM2.5 pollution sources.
Point 9: * Need reference for data in Figure 1 & Table 1.
Response 9: I think I have added relevant reference.
Point 10: *Lines 77-78: delete this statement, the same meaning as statement in lines 74-77.
Response 10: We are sorry. We have deleted this statement.
Point 11: *Need reference for the statement in lines 50-51 & lines 82-84 & lines 88-97 & lines 134-140 & lines 223-224.
Response 11: We have solved most of the problems after several rounds of revision. Because the time is not well planned, we are not sure whether the revision is thorough or not when it is about to be submitted. I hope you can give us another chance to further improve the article.
Point 12: *Lines 113-115: so what? What is the main findings and conclusions of Hu (2019)?
Response 12: Another reviewer suggested that lines 82-124, this part should also be significantly resized and moved in the Discussions section. We have deleted Lines 113-115 to meet his requirement.
Point 13: *Line 144: which study? Need reference.
Response 13: We provided the citation number at the end of this sentence.
Point 14: *Lines 151-155: grammatical errors, please rephrase.
Response 14: We have rephrased these sentences.
Point 15: *Please divide the Materials and Methods section to sub-sections (2.1. Study area, 2.2. simulation software).
Response 15: We have divided the Materials and Method section to sub-sections.
Point 16: *Lines 167-186: more suitable to be included to the introduction section.
Response 16: Another reviewer suggested us that the geographical description (Lines 157-186) of the selected site should be drastically reduced, or even better eliminated. We have drastically reduced this part and moved relevant information to the introduction part.
Point 17: *You did not refer to Figure 2 in the text.
Response 17: We have referred to figure 2. We have Point ed out 'According to the specific information of this road section, the researchers first built a three-dimensional model on sketch up 2021, as shown in Figure 1. The researchers used the INX plug-in in sketch up to assign material information to buildings and grounds in the software and convert the model into the one that can be used in ENVI-met software. ’’
Point 18: *Figure 2, need to include more data: north direction, scale, grid line, etc.
Response 18: Another reviewer suggested us that Figure 2 should be removed or replaced with a more suitable figure, for example a three-dimensional representation of the model, rather than an aerial photo. We have used a three-dimensional representation of the model to replace the aerial photo.
Point 19: *Line 191: Bruse et al., could not find it in the references list.
Response 19: We have added the reference.
Point 20: *Line 202: do you mean Figure 2? *Line 220: do you mean Table 1? *Line 245: do you mean Figure 2?
Response 20: We are sorry. We have corrected the error. We think the figure number in the text correspond with the figure which we refer to.
Point 21: *Please add a table at the end of Materials and Methods section that include list of inputs data you entered to the model for each scenario.
Response 21: We have added some tables about the inputs data. We are sorry it is only 15 minutes left for me to resubmit the article. I am not able to put all relevant data at the end of Materials and Methods section. If you give us chance to revise the article again. Could you also tell us do we need to add more relevant information and put the table at the end of Materials and Methods section.
Point 22: *Lines 254-258: be consistent when using units ug and μm?
Response 22: We are sorry for the slip of pen. The unit should be μg/m3.
Point 23: *Lines 273-275: need evidence, reference.
Response 23: We are sorry. We have provided relevant evidence and reference.
Point 24: *Need to see some results in the conclusion section. I have notice repetition for some information in the conclusion section.
Response 24: We have done the research again. We have rewritten the conclusion section. We have tried our best to avoid repetition.
Abstract of the article
With the rapid development of cities and the rapid increase in automobile ownership, traffic has become one of the main sources of PM2.5 pollution, which can be reduced by road greening through sedimentation, blocking, adhesion, and absorption. Using the method of combining field monitroing and ENVI-met simulation, the influence of the tree height on the PM2.5 concentration on both sides of the city streets was discussed. The influence of tree height on PM2.5 under five conditions was analyzed, including 10-meter-tall trees (i), 15-meter-tall trees (ii), alternating distribution of 15-meter-tall and 10 -meter-tall trees (iii), 5-meter-tall trees (IV), no trees on either side of the road (V). The results show that: Roadside trees can increase the concentration of PM2.5 in the narrow space of street canyons. However, without roadside trees, PM2.5 from traffic sources is not reduced in time, it is more easily spread to the distance. The height of the roadside trees is 5 meters, and their crown widths are smaller than those of other trees, and there is a relatively wide space between those trees. Compared with the higher roadside tree models with larger crown widths, the concentration of PM2.5 on the roadway and the downwind sidewalk is relatively low. In the three models (i- iii) with tree height above or equal to 10 meters, the PM2.5 concentration around the trees do not show regular change with the change of tree height. Due to the tree height of 10 meters and 15 meters, the crown width is larger, and the alternate distribution of tree height of 15 meters and 10 meters fails to make the PM2.5 concentration in the models lower than that in the models with tree height of 15 meters or 10 meters. The reasonable height of roadside trees in street canyons helps improve the wind circulation to promote the diffusion of PM2.5 pollution. There’s no optimal height of roadside trees for PM2.5 pollution in street canyons, so it is necessary to select the height reasonably according to the specific situation.
Round 2
Reviewer 2 Report
Authors replied to all the points previously revised, effectively making changes. I personally have no other changes to suggest. Best regards.
Reviewer 3 Report
Thanks for addressing my comments.